# Associations of diarised sleep onset time, period and duration with total and central adiposity in a biethnic sample of young children: the Born in Bradford observational cohort study

Paul James Collings ![ORCID],[1,2] Jane Elizabeth Blackwell,[1,3] Elizabeth Pal,[4] Helen L Ball,[5] John Wright ![ORCID] [2]

¹Department of Health Sciences, University of York, York, UK
²Bradford Institute for Health Research, Bradford, UK
³Leeds and York Partnership NHS Foundation Trust, Leeds, UK
⁴Better Start Bradford, Bradford Trident, Bradford, UK
⁵Department of Anthropology, Durham Infancy and Sleep Centre, Durham University, Durham, UK

**Correspondence to**
Dr Paul James Collings;
paul.collings@york.ac.uk

## ABSTRACT

**Objectives** To investigate associations of parent-reported sleep characteristics with adiposity levels in a biethnic sample of young children.

**Design** A cross-sectional observational study.

**Setting** The Born in Bradford 1000 study, UK.

**Participants** Children aged approximately 18 months (n=209; 40.2% South Asian; 59.8% white) and 36 months (n=162; 40.7% South Asian; 59.3% white).

**Primary and secondary outcome measures** Children's body mass index (BMI) z-score, sum of two-skinfolds (triceps and subscapular) and waist circumference. Adjusted regression was used to quantify associations of sleep parameters with adiposity stratified by ethnicity and age group. The results are beta coefficients (95% CIs) and unless otherwise stated represent the difference in outcomes for every 1-hour difference in sleep parameters.

**Results** The average sleep onset time was markedly later in South Asian (21:26±68 min) than white children (19:41±48 min). Later sleep onset was associated with lower BMI z-score (−0.3 (−0.5 to −0.0)) and sum of two-skinfolds (−1.5 mm (−2.8 mm to −0.2 mm)) in white children aged 18 months and higher BMI z-score in South Asian children aged 36 months (0.3 (0.0–0.5)). Longer sleep duration on weekends than weekdays was associated with higher BMI z-score (0.4 (0.1–0.8)) and waist circumference (1.2 cm (0.3–2.2 cm)) in South Asian children aged 18 months, and later sleep onset on weekends than weekdays was associated with larger sum of two-skinfolds (1.7 mm (0.3–3.1 mm)) and waist circumference (1.8 cm (0.6–2.9 cm)). Going to sleep ≥20 min later on weekends than weekdays was associated with lower waist circumference in white children aged 18 months (−1.7 cm (−3.2 cm to −0.1 cm)).

**Conclusions** Sleep timing is associated with total and central adiposity in young children but associations differ by age group and ethnicity. Sleep onset times and regular sleep schedules may be important for obesity prevention.

## Strengths and limitations of this study

► This is the first study to investigate myriad sleep dimensions and how they relate to direct markers of adiposity in young children.

► It is also the first study to investigate differences in these associations between UK born South Asian and white children living in a deprived urban setting.

► Sleep parameters were mutually adjusted for each other, and the analyses were adjusted for a broad range of covariates including potential mediating factors.

► Residual confounding and mediating effects may have persisted because of measurement imprecision in parent-reported covariates, and directions of association between exposures and outcomes cannot be inferred from this cross-sectional study.

► The proportion of parents who completed sleep diaries was low, meaning the study sample is susceptible to selection biases, and when stratified by ethnicity and age some groups were small.

regardless of waking intervals) with adiposity in school-aged children and youth.[1 2] Comparatively few studies have been conducted in infants and toddlers[3 4] despite sleep problems and sleep loss being prevalent in children aged <36 months.[5] Beyond sleep duration, it is beginning to be recognised that sleep is a multidimensional construct of partly overlapping dimensions, and that parameters such as sleep timing may also be influential with regard to adiposity level and obesity risk.[6]

Aspects of sleep timing include bedtimes, time of sleep onset, and variability in sleeping patterns such as differences between weekday and weekend sleep schedules. Preliminary evidence indicates that later bedtimes are associated with higher weight status in preschoolers[7–10] and primary school-aged children,[11] and that later sleep timing on

## INTRODUCTION

Many studies have investigated associations of sleep duration or the sleep period (elapsed time between sleep onset and termination

weekends than weekdays is correlated with higher risk of overweight and obesity.[12] Hypothesised mechanisms include circadian dysrhythmia causing hormone-mediated preferences for unhealthy calorific foodstuffs,[13 14] the effect of which may be compounded by later sleep onset enabling more time to eat, more opportunity for screen time,[15] and lower daily physical activity due to delayed waking and fatigue.[16] More studies are needed to investigate the independent associations of myriad sleep parameters with childhood adiposity and to ascertain the underlying mechanisms of action. A research focus on young children is important, to determine when associations between sleep parameters with adiposity first emerge, and to provide information about how to intervene by identifying modifiable target behaviours.[6]

Early childhood is a critical window for obesity prevention because it is a period when rapid weight gain occurs and behavioural patterns are first established.[17 18] Although an increasing number of studies have investigated sleep duration,[3 4] few have investigated associations of sleep timing or variability with adiposity in early childhood,[19–21] and additional investigation of ethnic minority and socioeconomically impoverished populations is needed. Children of non-white ethnicity and children from deprived backgrounds have unfavourable sleeping patterns and are at high risk for early onset obesity.[22 23] We have previously shown that UK South Asian children who live in a deprived urban setting go to bed much later than white children from the same area.[24] It is important to determine if shifted sleep schedules may in part explain why obesity rates are higher in South Asian than white children in the UK,[25] and why South Asian school-aged children have more centrally stored adiposity, which is metabolically harmful.[26] This information could help to reduce ethnic disparities in disease risk by facilitating behaviour change interventions and policy guidelines that are tailored to the needs of specific ethnic groups.

This study examined independent associations of a broad range of sleep parameters, including diarised sleep onset time, period and duration, and weekday to weekend variations in sleep parameters, with total and central adiposity in a young biethnic sample of UK children from a deprived urban setting.

## METHODS

This study was conducted in Bradford, which is the fifth largest local authority in England and one of the most deprived and ethnically diverse cities in the UK.[27] Bradford has an overweight and obesity prevalence of 21.8% in 4–5-year-old children and 38.4% in 10–11 year olds.[28] The Born in Bradford (BiB) 1000 study,[29] nested within the larger BiB pregnancy cohort,[30] aims to investigate modifiable risk factors for childhood obesity. Pregnant women (n=1916) were invited to the BiB 1000 study when they attended routine hospital appointments, 90.6% accepted the invitation (n=1735). Consent to medical records access was provided and periodic postnatal

assessments were carried out when the women's offspring were approximately 6, 12, 18, 24 and 36 months old. Parents who participated when children were aged about 18 months (n=1228; 70.7% of all BiB 1000 participants) or 36 months (n=1232; 71%) were asked to complete a sleep diary for their child. At each timepoint, diaries were completed for approximately 15% of all BiB 1000 participants (18 months: n=276; 36 months: n=262). The diary data were collected between October 2010 and September 2012. This complete-case analysis included only children with sleep diary data, concurrent adiposity measurements and information about potentially important covariates. Maternal ethnicity was used as a proxy for child ethnicity. Children with a mother belonging to an ethnic group other than South Asian (Pakistani/Indian/Bangladeshi) or white (British/other) were excluded from the study due to small numbers. All children were born in the UK (most children of South Asian heritage had Pakistani origin mothers (85%) and most white children had white British mothers (94%)). For brevity, we refer to children as being of South Asian or white ethnicity. The final sample included 209 children aged 18 months (12% of all BiB 1000 participants) and 162 children aged 36 months (9.3%).

### Sleep parameters

Parents completed sleep diaries, providing free-text responses about the time their child fell asleep on an evening and woke-up the next morning. The sleep period was calculated as elapsed time between sleep onset and waking the next day. Because the sleep period does not account for periods of overnight waking, which can be common in young children,[31] parents also reported their child's overnight sleep duration. Parents completed diaries on 2 weekdays and a weekend day. Mean weekday values were calculated by averaging data for the 2 weekdays; thereafter, daily averages over the course of a week were calculated using weekday to weekend weighting (in the ratio of 5:2). Differences in sleep parameters between weekdays and weekends were calculated by subtraction (weekend–weekday values).

### Adiposity markers

Child weight and height were assessed by trained researchers and body mass index (BMI) (kg/m$^2$) and z-scores were calculated.[32] Waist circumference was measured at the level of the navel. Skinfolds of the left triceps and subscapular were measured to the nearest 0.1 mm and summed. Reliability metrics indicated good intraobserver and interobserver technical error for measurements.[33]

### Covariates

Potential confounders and mediators of associations were selected based on previous evidence linking them with sleep and obesity.[17 22] A multidimensional marker of socioeconomic status was synthesised from information collated during interviews with parents (usually

mothers) about their education, employment, housing tenure, financial situation and ownership of goods. For the purposes of this study, from an initial five categories,[34] children were classified as belonging to one of three parental socioeconomic status groups: least deprived (least socioeconomically deprived and most educated), moderately deprived (employed and not materially deprived/employed but no access to money) and most deprived (benefits but not materially deprived/most economically deprived). Maternal age was self-reported in pregnancy, and the number of previous births (used to group mothers as primaparous (pregnant for the first time) or not), child gender, gestational age and birth weight were all extracted from medical records. Maternal height and weight were measured at the 18 months and 36 months timepoints and were used to calculate maternal BMI. Parents reported whether or not their child napped on sleep diary days; these data were used to create three categories of napping frequency (never: napped zero days; occasionally: napped 1 or 2 days; every day: napped all 3 days). On each sleep diary reporting day, parents also provided free-text responses about the time their child ate their last meal of the day. Mean weekday values were calculated by averaging data across the 2 weekdays, and weekday to weekend weighting was used to calculate the average daily time that the child ate their last meal; differences between weekdays and weekends were calculated by subtraction. Infant dietary data were collected using a validated food frequency questionnaire that was modified to include ethnic-specific foodstuffs. Dietary constructs indicative of unhealthy snacking (frequency of biscuit, crisps, cakes, sweets, chocolate and sugar-sweetened beverage consumption) and fruit and vegetable consumption in the previous 4–12 weeks were derived.[24] Parents reported the number of hours their child watched TV on a typical weekday and weekend day; weighting was used to calculate average daily TV viewing and weekday to weekend differences were calculated.[35] When children were aged 36 months, a parent-reported Early Years Physical Activity Questionnaire was used to estimate children's physical activity level.[36] Calendar dates of the 18 months and 36 months assessments were recorded and categorised by season (summer/autumn or spring/winter).

### Patient and public involvement

The BiB research team regularly convenes a Parent Governors' patient and public involvement (PPI) group, BiB participants who are now aged 10–12 years can become Young Ambassadors of the study and BiB runs regular community events and science festivals (https://bornin-bradford.nhs.uk/news-events/events/). These committees and events help to shape our research by allowing us to learn about the opinions, concerns and ideas of the community. A study author (JEB) also recently chaired a PPI event that was held in conjunction with the Sleep Charity (https://thesleepcharity.org.uk/). Attendees were parents of children with sleep difficulties, staff who work in residential settings with children, sleep

practitioners and representatives from parent and carer forums. Emerging themes from the session included: (1) there is a lack of information about assisting development of healthy sleep routines in infants, (2) more information and knowledge about the consequences of poor sleep in infants is needed and (3) early intervention is vital and parents of young children should be offered evidence-based sleep interventions for their child.

### Statistics

Correlations between individual sleep parameters and adiposity markers were calculated using Pearson or Spearman methods as appropriate.[37] Adjusted linear regression was used to investigate cross-sectional associations between sleep parameters with adiposity. Model 1 adjusted associations for sex, age, socioeconomic status, maternal pregnancy age, parity, gestational age, birth weight, maternal BMI, season of measurement and napping frequency. Model 2 further adjusted for potential confounding or mediating factors, including unhealthy snacking, fruit and vegetable intake, and TV viewing. Model 3 mutually adjusted sleep period and duration variables for sleep onset time, and vice versa. All analyses were performed and stratified by ethnic group and age group. Interaction effects between the sleep period and duration with sleep onset time were examined by introducing multiplicative terms to models (eg, sleep onset×duration). Non-linearity of associations was examined by adding quadratic terms for sleep parameters. There was some evidence for nonlinear associations between average daily sleep onset and weekday to weekend differences in sleep onset with waist circumference and sum of two-skinfolds; hence, sleep onset variables were additionally trichotomised and adjusted linear regression was used to estimate group marginal means (and 95% CIs) for these outcomes. Data for the sum of two-skinfolds were slightly skewed, but because results were consistent regardless of whether the data were log-transformed or not, results based on the untransformed data are presented. Sensitivity analyses further adjusted associations for the time of last meal and physical activity (not applicable to 18 months models when physical activity was not assessed), potentially important covariates but missing data; models with waist circumference or sum of two-skinfolds as dependent variables were further adjusted for height. Analyses were performed with Stata/SE V.16.1 software (StataCorp, College Station, Texas, USA). Variance inflation factors were calculated to highlight potential multicollinearity and $p < 0.05$ was deemed to indicate statistically significant associations, but all results are interpreted with emphasis on the range of plausible values of associations as shown by CIs.[38]

## RESULTS
### Sample characteristics
Participant details are listed in table 1. Child ages ranged from 16.4 months to 22.8 months and 35.5 months to 39.5 months at each respective timepoint. The average daily sleep onset time was consistently after 21:00 in South

**Table 1** Description of study participants

| | South Asian | | White | |
|---|---|---|---|---|
| | 18 months (n=84) | 36 months (n=66) | 18 months (n=125) | 36 months (n=96) |
| Sex (n (%) boys) | 43 (51.2) | 32 (48.5) | 58 (46.4) | 38 (39.6) |
| Age (months) | 18.1 (0.7) | 36.7 (1.1) | 18.3 (1.1) | 36.5 (0.5) |
| Parental socioeconomic status (n (%)) | | | | |
| Least deprived | 22 (26.2) | 15 (22.7) | 32 (25.6) | 30 (31.3) |
| Moderately deprived | 29 (34.5) | 20 (30.3) | 70 (56.0) | 53 (55.2) |
| Most deprived | 33 (39.3) | 31 (47.0) | 23 (18.4) | 13 (13.5) |
| Maternal ethnicity (n (%) Pakistani origin) | 73 (86.9) | 55 (83.3) | – | – |
| Maternal ethnicity (n (%) white British) | – | – | 119 (95.2) | 89 (92.7) |
| Maternal pregnancy age (years) | 27.6 (7.3) | 30.6 (7.6) | 29.1 (10.3) | 28.8 (8.2) |
| Parity (n (%) primaparous) | 30 (35.7) | 21 (31.8) | 65 (52.0) | 60 (62.5) |
| Birth weight (g) | 3156±465 | 3066±552 | 3400±543 | 3358±563 |
| Gestational age (weeks) | 39.6±1.6 | 39.4±1.6 | 39.8±1.5 | 39.6±1.9 |
| Season (n (%) summer/autumn) | 71 (84.5) | 22 (33.3) | 108 (86.4) | 18 (18.8) |
| Unhealthy snacks (n/week) | 10 (10) | 17 (23) | 9 (8) | 11 (12) |
| Fruit and vegetables (portions/day) | 6±2 | 6±3 | 5±2 | 5±3 |
| Physical activity (hours/day) | – | 2.8±1.5 | – | 2.7 (2.0) |
| Maternal BMI (kg/m$^2$) | 26.4 (6.3) | 27.2 (6.6) | 25.1 (8.0) | 25.8 (7.0) |
| Napping frequency (n (%)) | | | | |
| Never | 0 (0) | 24 (36.4) | 4 (3.2) | 47 (49.0) |
| Occasionally | 7 (8.3) | 26 (39.4) | 20 (16.0) | 44 (45.8) |
| Everyday | 77 (91.7) | 16 (24.2) | 101 (80.8) | 5 (5.2) |
| Weighted daily average | | | | |
| TV viewing (hours/day) | 1.0 (1.6) | 1.5 (1.7) | 0.6 (1.0) | 1.5 (1.1) |
| Time of last meal (pm) | 19:23±66 min | 19:20±59 min | 17:28±51 min | 17:32±43 min |
| Sleep onset (pm) | 21:36±69 min | 21:12±63 min | 19:50±50 min | 19:54±44 min |
| Sleep offset (am) | 08:21±63 min | 08:23±63 min | 07:11±47 min | 07:12±50 min |
| Sleep period (hours/night) | 10.8±1.0 | 11.2±0.8 | 11.4±0.8 | 11.3±0.7 |
| Sleep duration (hours/night) | 10.5±1.0 | 11.0±0.9 | 11.1±1.1 | 11.3±0.8 |
| Weekday to weekend difference | | | | |
| TV viewing (hours/day) | −0.1±0.6 | 0.0±0.6 | 0.0±0.6 | 0.0±0.9 |
| Time of last meal (hours) | 0.2±0.7 | 0.2±0.7 | 0.1±0.8 | 0.2±0.7 |
| Sleep onset (hours) | 0.1±0.6 | 0.3±0.8 | 0.1±0.9 | 0.1±0.8 |
| Sleep offset (hours) | 0.2±0.9 | 0.3±0.8 | 0.3±0.8 | 0.2±0.9 |
| Sleep period (hours/night) | 0.1±1.0 | 0.0±0.8 | 0.2±1.2 | 0.1±1.0 |
| Sleep duration (hours/night) | 0.0±0.9 | −0.1±1.0 | 0.0±1.0 | 0.0±1.1 |
| BMI (kg/m$^2$) | 16.0±1.3 | 15.8±1.2 | 16.7±1.2 | 16.4±1.2 |
| BMI z-score | −0.8±1.1 | −0.2±1.0 | −0.2±0.9 | 0.3±0.8 |
| Waist circumference (cm) | 44.9±2.8 | 49.2±3.1 | 46.3±3.0 | 50.1±3.1 |
| Sum of two-skinfolds (mm) | 17.1 (3.8) | 15.8 (3.4) | 17.4 (4.3) | 17.2 (4.8) |

Normally distributed continuous variables are described as mean±SD. Time of last meal, sleep onset and offset are described as clock times±SD in minutes. Non-normally distributed continuous variables are described as median (IQR). Positive weekday to weekend differences indicate more TV viewing or sleep at the weekend than on weekdays, or later clock times at the weekend. There were missing data: physical activity available for n=60 South Asian children at 36 months and n=89 white children at 36 months; time of last meal available for n=71 South Asian children at 18 months, n=59 South Asian children at 36 months, n=119 white children at 18 months, n=91 white children at 36 months; sleep duration available for n=63 South Asian children at 18 months, n=45 South Asian children at 36 months, n=108 white children at 18 months, n=90 white children at 36 months; waist circumference available for n=77 South Asian children at 18 months, n=60 South Asian children at 36 months, n=109 white children at 18 months, n=96 white children at 36 months; sum of two-skinfolds available for n=64 South Asian children at 18 months, n=38 South Asian children at 36 months, n=84 white children at 18 months, n=67 white children at 36 months.
BMI, body mass index.

Asian children and before 20:00 in white children. Sleep onset time was inversely correlated with the sleep period (South Asian children aged 18 months: −0.52; South Asian children aged 36 months: −0.38; white children aged 18 months: −0.52; white children aged 36 months: −0.37) and to a lesser extent sleep duration (South Asian children aged 18 months: −0.42; South Asian children aged 36 months: −0.10; white children aged 18 months: −0.38; white children aged 36 months: −0.27). Correlations between the sleep period and sleep duration were weaker in South Asian than white children (South Asian children aged 18 months: 0.72; South Asian children aged 36 months: 0.66; white children aged 18 months: 0.81; white children aged 36 months: 0.83). Correlations between adiposity markers differed by ethnicity and age group, but consistently BMI z-score was more strongly correlated with waist circumference (South Asian children aged 18 months: 0.69; South Asian children aged 36 months: 0.71; white children aged 18 months: 0.61; white children aged 36 months: 0.79) than the sum of two-skinfolds (South Asian children aged 18 months: 0.54; South Asian children aged 36 months: 0.28; white children aged 18 months: 0.44; white children aged 36 months: 0.53); correlations between waist circumference and the sum of two-skinfolds were weakest (South Asian children aged 18 months: 0.47; South Asian children aged 36 months: 0.19; white children aged 18 months: 0.26; white children aged 36 months: 0.47).

### South Asian children
Table 2 shows that in South Asian children aged 18 months there were no significant associations between average daily sleep parameters with adiposity. In children aged 36 months, there was some evidence in model 3 that later daily sleep onset was associated with higher BMI z-score; the association attenuated when further adjusted for final mealtime (β=0.2 (95% CI: −0.2 to 0.6), p=0.34). There were no other significant associations for average daily sleep parameters. Figure 1 presents the group marginal means for waist circumference and the sum of two-skinfolds stratified by sleep onset categories; there were no group differences. With regard to weekday to weekend differences, table 2 shows that longer sleep duration on weekends than weekdays was consistently associated with higher adiposity in South Asian children aged 18 months, and in the same group of children, later sleep onset on weekends than weekdays was associated with larger waist circumference and sum of two-skinfolds. In South Asian children aged 36 months, later sleep onset on weekends than weekdays was associated with larger sum of two-skinfolds in model 2, but the association attenuated when further adjusted for the sleep period. Figure 2 shows the adjusted marginal means for waist circumference and sum of two-skinfolds stratified by three categories of weekday to weekend differences in sleep onset. There was some evidence that compared to children with consistent sleep onset times (±20 min of each other), children aged 18 months who went to sleep ≥20 min later on weekends than weekdays had larger waist circumferences (1.5 cm (−0.1

cm to 3.0 cm), p=0.060) and sum of two-skinfolds (1.9 mm (−0.2 mm to 4.1 mm), p=0.078). Similarly, there was some indication that aged 36 months, children who went to sleep ≥20 min later on weekends than weekdays had larger sum of two-skinfolds (2.0 mm (−0.2 mm to 4.3 mm), p=0.074).

### White children
Table 3 shows that for white children aged 36 months there were no significant associations, although there was some indication that later sleep onset on weekends than weekdays was associated with smaller sum of two-skinfolds. In children aged 18 months, independent of all covariables, including sleep period and duration, later daily sleep onset was associated with smaller sum of two-skinfolds and lower BMI z-score (though further adjustment for final mealtime attenuated the latter association (−0.2 (−0.5 to 0.2), p=0.28). Figure 1 illustrates that at age 18 months, compared to children with a sleep onset time of 19:30 or earlier, children who began to sleep between 19:30 and 20:30 had smaller waist circumferences (−1.5 cm (−2.9 cm to −0.1 cm), p=0.035). Figure 2 highlights that children aged 18 months who went to sleep ≥20 min later on weekends than weekdays had smaller waist circumferences than children with consistent sleep onset times (−1.7 cm (−3.2 cm to −0.1 cm), p=0.038).

### Sensitivity analyses
Further adjustment for physical activity in models that included children aged 36 months, and adjustment for height when waist circumference and the sum of two-skinfolds were modelled as dependent variables, did not influence any of the reported associations. There was no evidence for interaction effects between sleep onset with the sleep period or sleep duration. Variance inflation factors were within acceptable limits (<10; the majority were <2.5).

### DISCUSSION
This study investigated associations of diarised sleep onset time, period and duration, and weekday to weekend differences in sleep parameters, with total and central adiposity in young children aged 18 months and 36 months. We discovered age-specific and ethnicity-specific associations for sleep onset times, which we in part attribute to markedly different sleep schedules between ethnic groups; South Asian children went to sleep on average nearly 2 hours later than white children. Later average sleep onset and later sleep timing on weekends than weekdays were associated with higher adiposity in South Asian children, and conversely with lower adiposity in white children, particularly those aged 18 months. Longer sleep durations on weekends than weekdays were associated with higher adiposity in South Asian children aged 18 months.

Numerous systematic reviews have summarised the evidence for associations of sleep duration or the sleep period with adiposity in children and adolescents.[1 2] Of note, one review meta-analysed the results of 6 prospective

**Table 2** Associations of sleep onset, period and duration with adiposity in South Asian children, stratified by age group

| | Body mass index z-score | | | | | Waist circumference (cm) | | | | | Sum of two-skinfolds (mm) | | | | |
|---|---|---|---|---|---|---|---|---|---|---|---|---|---|---|---|
| | | Weighted daily average | | Weekday to weekend difference | | | Weighted daily average | | Weekday to weekend difference | | | Weighted daily average | | Weekday to weekend difference | |
| | n | β (95% CI) | P value | β (95% CI) | P value | n | β (95% CI) | P value | β (95% CI) | P value | n | β (95% CI) | P value | β (95% CI) | P value |
| **18 months** | | | | | | | | | | | | | | | |
| **Model 1** | | | | | | | | | | | | | | | |
| Sleep onset | 84 | −0.1 (−0.3 to 0.2) | 0.66 | 0.1 (−0.4 to 0.5) | 0.76 | 77 | −0.4 (−1.0 to 0.2) | 0.24 | 0.8 (−0.3 to 1.8) | 0.17 | 64 | −0.2 (−1.1 to 0.7) | 0.68 | 1.3 (−0.1 to 2.8) | 0.070 |
| Sleep period | 84 | 0.2 (−0.1 to 0.4) | 0.16 | 0.0 (−0.3 to 0.3) | 0.84 | 77 | 0.3 (−0.3 to 1.0) | 0.30 | 0.0 (−0.7 to 0.8) | 0.91 | 64 | 0.2 (−0.7 to 1.0) | 0.70 | −0.3 (−1.4 to 0.8) | 0.56 |
| Sleep duration | 63 | 0.2 (−0.2 to 0.5) | 0.29 | 0.3 (−0.1 to 0.6) | 0.17 | 58 | 0.3 (−0.6 to 1.2) | 0.54 | 0.6 (−0.4 to 1.7) | 0.21 | 49 | 0.5 (−0.7 to 1.6) | 0.41 | 0.8 (−0.6 to 2.2) | 0.26 |
| **Model 2** | | | | | | | | | | | | | | | |
| Sleep onset | 84 | 0.0 (−0.2 to 0.2) | 0.93 | 0.0 (−0.4 to 0.4) | 0.93 | 77 | −0.3 (−0.9 to 0.4) | 0.41 | 0.7 (−0.4 to 1.8) | 0.21 | 64 | −0.1 (−0.9 to 0.8) | 0.90 | **1.6 (0.1 to 3.1)** | **0.036** |
| Sleep period | 84 | 0.2 (−0.1 to 0.4) | 0.22 | 0.1 (−0.2 to 0.4) | 0.59 | 77 | 0.3 (−0.4 to 1.0) | 0.39 | 0.3 (−0.5 to 1.1) | 0.47 | 64 | 0.1 (−0.8 to 1.0) | 0.89 | −0.2 (−1.4 to 0.9) | 0.72 |
| Sleep duration | 63 | 0.2 (−0.1 to 0.5) | 0.27 | 0.3 (−0.0 to 0.7) | 0.066 | 58 | 0.3 (−0.6 to 1.2) | 0.55 | 0.9 (−0.1 to 1.9) | 0.075 | 49 | 0.3 (−0.9 to 1.5) | 0.61 | 0.9 (−0.6 to 2.3) | 0.23 |
| **Model 3** | | | | | | | | | | | | | | | |
| Sleep onset* | 84 | 0.1 (−0.2 to 0.4) | 0.4 | 0.0 (−0.5 to 0.5) | 0.99 | 77 | −0.2 (−0.9 to 0.6) | 0.65 | 1.2 (−0.1 to 2.4) | 0.076 | 64 | 0.0 (−1.1 to 1.1) | 0.96 | **1.8 (0.1 to 3.5)** | **0.034** |
| Sleep onset† | 63 | 0.2 (−0.1 to 0.5) | 0.23 | 0.4 (−0.1 to 0.8) | 0.12 | 58 | 0.2 (−0.6 to 1.1) | 0.58 | **1.8 (0.6 to 2.9)** | **0.004** | 49 | 0.0 (−1.0 to 1.1) | 0.92 | **1.7 (0.3 to 3.1)** | **0.022** |
| Sleep period | 84 | 0.2 (−0.1 to 0.5) | 0.14 | 0.1 (−0.2 to 0.4) | 0.63 | 77 | 0.2 (−0.6 to 1.0) | 0.60 | 0.7 (−0.2 to 1.5) | 0.14 | 64 | 0.0 (−1.1 to 1.1) | 0.93 | 0.3 (−0.9 to 1.5) | 0.61 |
| Sleep duration | 63 | 0.3 (−0.1 to 0.6) | 0.14 | **0.4 (0.1 to 0.8)** | **0.024** | 58 | 0.3 (−0.6 to 1.3) | 0.46 | **1.2 (0.3 to 2.2)** | **0.012** | 49 | 0.3 (−0.9 to 1.5) | 0.61 | 1.4 (−0.0 to 2.9) | 0.056 |
| **36 months** | | | | | | | | | | | | | | | |
| **Model 1** | | | | | | | | | | | | | | | |
| Sleep onset | 66 | 0.1 (−0.1 to 0.4) | 0.17 | −0.3 (−0.6 to 0.0) | 0.085 | 60 | −0.1 (−1.0 to 0.7) | 0.74 | **−1.2 (−2.4 to −0.0)** | **0.048** | 38 | −0.3 (−1.3 to 0.7) | 0.52 | 0.9 (−0.3 to 2.1) | 0.14 |
| Sleep period | 66 | 0.1 (−0.2 to 0.4) | 0.50 | 0.2 (−0.1 to 0.5) | 0.14 | 60 | −0.4 (−1.7 to 0.8) | 0.48 | **1.1 (0.0 to 2.3)** | **0.047** | 38 | 0.0 (−1.2 to 1.3) | 0.96 | −0.3 (−1.8 to 1.1) | 0.64 |
| Sleep duration | 45 | 0.2 (−0.1 to 0.5) | 0.15 | −0.0 (−0.4 to 0.3) | 0.82 | 42 | 0.5 (−0.6 to 1.5) | 0.35 | 0.3 (−1.0 to 1.6) | 0.63 | 26 | 0.2 (−0.9 to 1.2) | 0.71 | −0.1 (−1.8 to 1.6) | 0.92 |
| **Model 2** | | | | | | | | | | | | | | | |
| Sleep onset | 66 | 0.2 (−0.1 to 0.4) | 0.13 | −0.2 (−0.6 to 0.1) | 0.15 | 60 | 0.1 (−0.8 to 1.0) | 0.86 | −1.0 (−2.2 to 0.2) | 0.11 | 38 | −0.2 (−1.4 to 1.0) | 0.75 | **1.3 (0.1 to 2.5)** | **0.033** |
| Sleep period | 66 | 0.1 (−0.2 to 0.5) | 0.42 | 0.2 (−0.1 to 0.5) | 0.14 | 60 | −0.4 (−1.7 to 0.9) | 0.55 | **1.2 (0.1 to 2.3)** | **0.032** | 38 | 0.1 (−1.2 to 1.4) | 0.84 | −1.1 (−2.7 to 0.4) | 0.15 |
| Sleep duration | 45 | 0.2 (−0.1 to 0.5) | 0.27 | 0.1 (−0.3 to 0.5) | 0.67 | 42 | 0.3 (−0.8 to 1.4) | 0.55 | 0.5 (−0.8 to 1.9) | 0.44 | 26 | −0.0 (−1.1 to 1.0) | 0.97 | −0.8 (−3.1 to 1.5) | 0.43 |
| **Model 3** | | | | | | | | | | | | | | | |
| Sleep onset* | 66 | **0.3 (0.0 to 0.5)** | **0.042** | −0.2 (−0.6 to 0.2) | 0.27 | 60 | 0.0 (−0.9 to 0.9) | 0.97 | −0.2 (−1.7 to 1.3) | 0.80 | 38 | −0.2 (−1.5 to 1.1) | 0.79 | 1.2 (−0.3 to 2.8) | 0.12 |
| Sleep onset† | 45 | 0.1 (−0.1 to 0.4) | 0.34 | −0.2 (−0.8 to 0.3) | 0.43 | 42 | −0.3 (−1.2 to 0.6) | 0.53 | 0.0 (−1.9 to 2.0) | 0.95 | 26 | −0.8 (−2.2 to 0.7) | 0.25 | 0.4 (−2.3 to 3.1) | 0.72 |
| Sleep period | 66 | 0.3 (−0.1 to 0.6) | 0.12 | 0.1 (−0.2 to 0.5) | 0.42 | 60 | −0.4 (−1.8 to 1.0) | 0.57 | 1.1 (−0.3 to 2.5) | 0.11 | 38 | 0.1 (−1.3 to 1.5) | 0.89 | −0.4 (−2.2 to 1.4) | 0.68 |
| Sleep duration | 45 | 0.2 (−0.1 to 0.5) | 0.23 | 0.0 (−0.4 to 0.4) | 0.85 | 42 | 0.3 (−0.8 to 1.4) | 0.55 | 0.5 (−0.9 to 2.0) | 0.45 | 26 | −0.1 (−1.2 to 0.9) | 0.80 | −0.2 (−3.4 to 2.9) | 0.85 |

Results are beta coefficients (95% CIs) and represent the difference in outcomes for every 1 hour later to sleep, 1 hour longer sleep period or duration on weekends than weekdays, or 1 hour later to sleep on weekends than weekdays. Model 1 was adjusted for sex, age, socioeconomic status, maternal pregnancy age, parity, gestational age, birth weight, maternal body mass index, season of measurement and napping frequency (models for weekday to weekend differences were further adjusted for the average weekday value of the exposure variable). Model 2 was further adjusted for unhealthy snacking, fruit and vegetable intake, and TV viewing (models for weekday to weekend sleep differences were adjusted for average weekday TV viewing and weekday to weekend differences in TV viewing). Model 3 further adjusted sleep period and duration variables for sleep onset, and vice versa (models for weekday to weekend differences were mutually adjusted for average weekday and weekday to weekend differences in the sleep period, duration or onset as appropriate). Statistically significant results (p<0.05) are highlighted bold. Sensitivity analyses confirmed that adjustment for physical activity (collected at 36 months only) did not confound model 3 results, but the association between weighted daily average sleep onset with body mass index z-score at 36 months was attenuated when adjusted for time of last meal (body mass index z-score (n=59): 0.2 (−0.2 to 0.6), p=0.34).
*Adjusted for the sleep period.
†Adjusted for sleep duration.

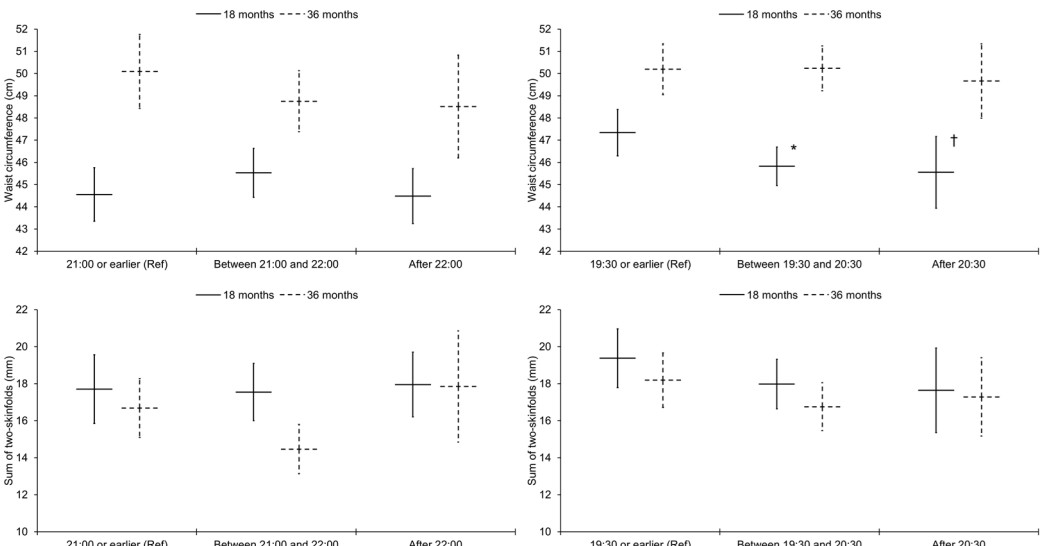

**Figure 1** Associations of sleep onset categories with waist circumference and sum of two-skinfolds, stratified by ethnicity and age group. Graphs on the left relate to South Asian children and graphs on the right to white children. Results are estimated marginal means (95% CIs) adjusted for sex, age, socioeconomic status, maternal pregnancy age, parity, gestational age, birth weight, maternal BMI, season of measurement, napping frequency, unhealthy snacking, fruit and vegetable intake, TV viewing and the sleep period. *Significantly different from the reference category of 19:30 or earlier (β=−1.5 (−2.9 to −0.1) cm, p=0.035); significant difference persisted when further adjusted for final mealtime (n=103: β=−1.7 (−3.2 to −0.1) cm, p=0.036). †Some evidence for a difference compared with the reference category: (β=−1.8 (−3.8 to 0.3) cm, p=0.088). Results are presented for South Asian children aged 18 months (21:00 or earlier: waist circumference, n=25; skinfolds, n=19; between 21:00 and 22:00: waist circumference, n=26; skinfolds, n=23; after 22:00: waist circumference, n=26; skinfolds, n=22) and 36 months (21:00 or earlier: waist circumference, n=22; skinfolds, n=16; between 21:00 and 22:00: waist circumference, n=28; skinfolds, n=17; after 22:00: waist circumference, n=10; skinfolds, n=5) and white children aged 18 months (19:30 or earlier: waist circumference, n=39; skinfolds, n=30; between 19:30 and 20:30: waist circumference, n=51; skinfolds, n=37; after 20:30: waist circumference, n=19; skinfolds, n=17) and 36 months (19:30 or earlier: waist circumference, n=36; skinfolds, n=25; between 19:30 and 20:30: waist circumference, n=42; skinfolds, n=29; after 20:30: waist circumference, n=18; skinfolds, n=13).

studies totalling 14 264 children aged <36 months. There was moderate heterogeneity in results, but each additional hour of sleep was associated with a negative change in BMI z-score over follow-up.[3] Limitations of the existing evidence include a preponderance of studies that have relied solely on BMI or BMI z-score as proxies for total adiposity. Studies have also considered few potential confounding or mediating factors and focused solely on sleep duration. Sleep is a multidimensional construct and it is important to unravel the independent relations of specific dimensions with adiposity. In a previous BiB 1000 study, we analysed repeated questionnaire data collected at four timepoints (12 months, 18 months, 24 months and 36 months) to quantify associations between parent-reported sleep duration with measured adiposity in 1338 UK South Asian and white children. Longer sleep duration predicted lower total and central adiposity but only in South Asian children.[24] Here, in a sub-sample of the same cohort, we were able to scrutinise and mutually adjust for myriad sleep parameters collected over the course of 3 days via parent-completed sleep diaries. Independent of sleep length, the average sleep onset time and variability in sleep behaviours between weekdays and weekends emerged as the key modifiable dimensions that predicted adiposity in South Asian and white children.

A recent review concluded that sleep timing, in particular later bedtimes, are associated with higher weight status in primary school-aged children.[11] Similarly, the few studies conducted thus far in 4–5 year olds indicate that later bedtimes, and most markedly bedtimes after 21:00, are associated with higher BMI z-score and obesity risk.[7–9] A particular study found that an association of short sleep with higher BMI z-score was only evident in children who went to bed after 21:00.[10] Just two studies have been performed in younger children. Zhang *et al* found no differences in internally derived BMI z-scores between early and late bedtime groups (separated using a crude median split at about 20:00) in a sample of children with ages ranging from 12 months to 26 months.[19] Roy *et al* found that later bedtimes in children aged 36 months were associated with higher odds of obesity, but the association attenuated when adjusted for sleep duration.[20] To the best of our knowledge, this is the first study to show that independent of the sleep period, later sleep onset is associated with higher BMI z-score in South Asian children aged 36 months. The association attenuated when adjusted for final mealtime, which provides some indication that later sleep onset may elevate obesity risk via later evening meals and higher caloric intake later in the day.[39] In line with this hypothesis, animal models and clinical trials have shown that early and time-restricted feeding has numerous physiological benefits, including lower adiposity.[40 41]

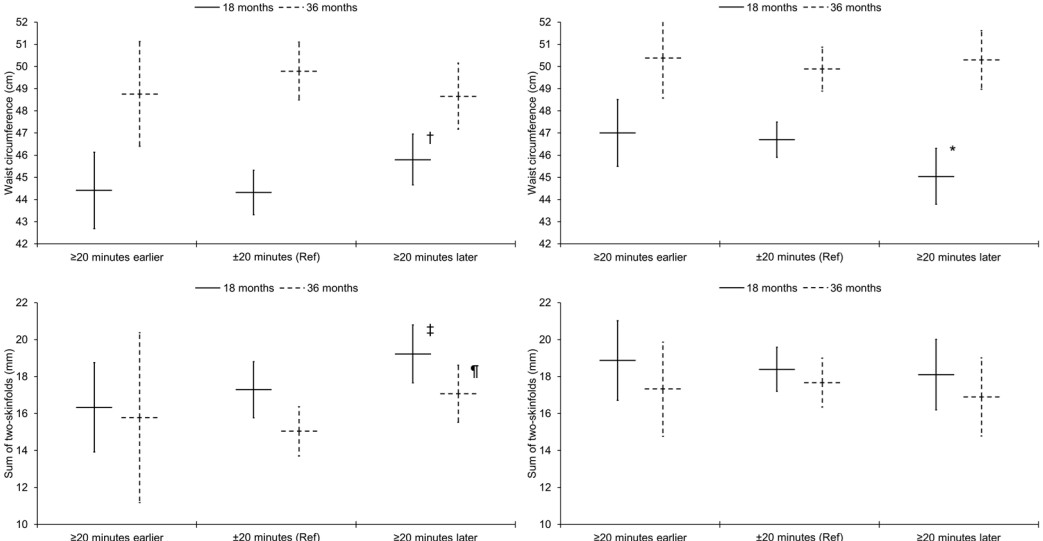

**Figure 2** Associations of weekday to weekend differences in sleep onset with waist circumference and sum of two-skinfolds, stratified by ethnicity and age group. Graphs on the left relate to South Asian children and graphs on the right to white children. Results are estimated marginal means (95% CIs) adjusted for sex, age, socioeconomic status, maternal pregnancy age, parity, gestational age, birth weight, maternal BMI, season of measurement, napping frequency, average weekday sleep onset, unhealthy snacking, fruit and vegetable intake, average weekday TV viewing, week to weekend differences in TV viewing, average weekday sleep period and week to weekend differences in the sleep period. *Significantly different from the reference category of ±20 min (β=−1.7 cm (−3.2 cm to −0.1 cm), p=0.038). †Some evidence for a difference compared with the reference category: (β=1.5 cm (−0.1 cm to 3.0 cm), p=0.060). ‡Some evidence for a difference compared with the reference category: (β=1.9 mm (−0.2 mm to 4.1 mm), p=0.078). ¶Some evidence for a difference compared with the reference category: (β=2.0 mm (−0.2 mm to 4.3 mm), p=0.074). Results are presented for South Asian children aged 18 months (≥20 min earlier: waist circumference, n=16; skinfolds, n=15; ±20 min: waist circumference, n=34; skinfolds, n=27; ≥20 min later: waist circumference, n=27; skinfolds, n=22) and 36 months (≥20 min earlier: waist circumference, n=10; skinfolds, n=3; ±20 min: waist circumference, n=28; skinfolds, n=21; ≥20 min later: waist circumference, n=22; skinfolds, n=14), and white children aged 18 months (≥20 min earlier: waist circumference, n=20; skinfolds, n=16; ±20 min: waist circumference, n=60; skinfolds, n=46; ≥20 min later: waist circumference, n=29; skinfolds, n=22) and 36 months (≥20 min earlier: waist circumference, n=19; skinfolds, n=14; ±20 min: waist circumference, n=47; skinfolds, n=35; ≥20 min later: waist circumference, n=30; skinfolds, n=18).

With regard to sleep variability, a pooled analysis of five studies conducted in school-aged children and youth indicated that later sleep timing on weekends than week-days was correlated with higher risk of overweight or obesity, although with a small effect size.[12] In toddlers, a recent study found that inconsistent sleep onset times over the course of a week predicted higher BMI z-scores, independent of device-measured physical activity and diet quality, and that inconsistent sleep indirectly explained an association between household poverty with higher BMI z-score.[21] We uniquely observed that later sleep timing on weekends than weekdays was associated with higher adiposity in South Asian children aged 18 months and larger sum of two-skinfolds in South Asian children aged 36 months, although the association in older children only approached statistical significance. The associations in children aged 18 months were independent of potential confounding or mediating factors, including TV viewing, unhealthy snacking, fruit and vegetable intake, and the final mealtime. It may be that these are not important mediators in very young children who are less autonomous and for whom a large part of their diet and screen time is governed by parents and carers. That said, we cannot exclude residual confounding or mediating effects as all covariate data were parent-reported and are

subject to random error and bias. Longer sleep duration on weekends than weekdays was associated with higher adiposity in South Asian children aged 18 months. This suggests that consistent sleep durations are important.

We observed that later daily sleep onset was associated with lower BMI z-score in white children aged 18 months. This observation may appear counterintuitive, but Roy *et al* similarly reported that independent of overnight sleep duration, each 1 hour later to bed was associated with −0.08 lower BMI z-score in 878 children aged 24 months, although their CI did narrowly cross the null of no association (95% CI: −0.17 to 0.01).[20] Following adjustment for final mealtime, our result for BMI z-score also attenuated to the null, but inverse associations with more direct adiposity indicators remained, including smaller sum of two-skinfolds and waist circumference as a function of later sleep onset. These unexpected results may partly be explained by early sleep onset obstructing time during early-to-mid evenings that could otherwise be spent physically active. Time-stamped accelerometer studies show that daily physical activity patterns are bimodal in 18 months olds. Physical activity levels first peak in the morning, slump after midday due to feeding and napping (81% of our white children aged 18 months napped every day), after which there is a second larger peak that begins

**Table 3** Associations of sleep onset, period and duration with adiposity in white children, stratified by age group

| | Body mass index z-score | | | | | Waist circumference (cm) | | | | | Sum of two-skinfolds (mm) | | | | |
|---|---|---|---|---|---|---|---|---|---|---|---|---|---|---|---|
| | Weighted daily average | | | Weekday to weekend difference | | Weighted daily average | | | Weekday to weekend difference | | Weighted daily average | | | Weekday to weekend difference | |
| | n | β (95% CI) | P value | β (95% CI) | P value | n | β (95% CI) | P value | β (95% CI) | P value | n | β (95% CI) | P value | β (95% CI) | P value |
| **18 months** | | | | | | | | | | | | | | | |
| **Model 1** | | | | | | | | | | | | | | | |
| Sleep onset | 125 | −0.2 (−0.4 to 0.0) | 0.086 | 0.0 (−0.2 to 0.2) | 0.70 | 109 | −0.5 (−1.2 to 0.2) | 0.18 | −0.6 (−1.4 to 0.2) | 0.17 | 84 | −1.0 (−2.1 to 0.0) | 0.061 | −0.1 (−1.2 to 1.0) | 0.88 |
| Sleep period | 125 | 0.0 (−0.2 to 0.2) | 0.83 | 0.1 (−0.1 to 0.2) | 0.44 | 109 | 0.2 (−0.6 to 0.9) | 0.64 | 0.1 (−0.5 to 0.7) | 0.70 | 84 | 0.1 (−1.0 to 1.2) | 0.82 | −0.1 (−0.9 to 0.8) | 0.86 |
| Sleep duration | 108 | 0.0 (−0.2 to 0.2) | 0.91 | 0.1 (−0.1 to 0.3) | 0.37 | 93 | 0.1 (−0.7 to 1.0) | 0.74 | 0.3 (−0.4 to 1.1) | 0.39 | 76 | −0.1 (−1.0 to 0.8) | 0.82 | 0.1 (−0.9 to 1.1) | 0.85 |
| **Model 2** | | | | | | | | | | | | | | | |
| Sleep onset | 125 | −0.2 (−0.4 to 0.0) | 0.11 | 0.1 (−0.1 to 0.3) | 0.39 | 109 | −0.5 (−1.2 to 0.2) | 0.18 | −0.5 (−1.3 to 0.3) | 0.23 | 84 | −1.0 (−2.1 to 0.0) | 0.060 | 0.1 (−1.0 to 1.2) | 0.85 |
| Sleep period | 125 | 0.0 (−0.2 to 0.2) | 0.92 | 0.0 (−0.1 to 0.2) | 0.81 | 109 | 0.1 (−0.6 to 0.9) | 0.71 | 0.0 (−0.6 to 0.6) | 0.99 | 84 | 0.1 (−1.0 to 1.1) | 0.91 | −0.3 (−1.2 to 0.5) | 0.45 |
| Sleep duration | 108 | 0.0 (−0.2 to 0.2) | 0.99 | 0.0 (−0.1 to 0.2) | 0.62 | 93 | 0.2 (−0.6 to 1.1) | 0.57 | 0.2 (−0.6 to 1.1) | 0.55 | 76 | −0.1 (−1.0 to 0.8) | 0.86 | −0.2 (−1.3 to 0.9) | 0.74 |
| **Model 3** | | | | | | | | | | | | | | | |
| Sleep onset* | 125 | −0.2 (−0.5 to 0.0) | 0.072 | 0.2 (−0.1 to 0.4) | 0.15 | 109 | −0.6 (−1.5 to 0.3) | 0.18 | −0.7 (−1.7 to 0.3) | 0.15 | 84 | **−1.5 (−2.8 to −0.2)** | **0.025** | −0.2 (−1.6 to 1.2) | 0.72 |
| Sleep onset† | 108 | **−0.3 (−0.5 to −0.0)** | **0.044** | 0.0 (−0.2 to 0.3) | 0.79 | 93 | −0.6 (−1.6 to 0.3) | 0.20 | −0.5 (−1.4 to 0.4) | 0.28 | 76 | **−1.5 (−2.9 to −0.0)** | **0.046** | −0.0 (−1.3 to 1.2) | 0.97 |
| Sleep period | 125 | −0.1 (−0.3 to 0.1) | 0.41 | 0.1 (−0.1 to 0.3) | 0.31 | 109 | −0.2 (−1.0 to 0.7) | 0.69 | −0.3 (−1.0 to 0.4) | 0.40 | 84 | −0.8 (−2.1 to 0.5) | 0.22 | −0.6 (−1.6 to 0.4) | 0.25 |
| Sleep duration | 108 | −0.1 (−0.2 to 0.1) | 0.49 | 0.1 (−0.1 to 0.3) | 0.48 | 93 | 0.0 (−0.9 to 0.9) | 0.99 | 0.1 (−0.8 to 0.9) | 0.88 | 76 | −0.6 (−1.6 to 0.4) | 0.23 | −0.3 (−1.5 to 0.8) | 0.58 |
| **36 months** | | | | | | | | | | | | | | | |
| **Model 1** | | | | | | | | | | | | | | | |
| Sleep onset | 96 | 0.0 (−0.2 to 0.3) | 0.71 | −0.1 (−0.3 to 0.1) | 0.32 | 96 | 0.2 (−0.7 to 1.1) | 0.63 | −0.1 (−0.9 to 0.8) | 0.89 | 67 | −0.3 (−1.6 to 0.9) | 0.57 | −0.6 (−1.6 to 0.5) | 0.29 |
| Sleep period | 96 | 0.0 (−0.3 to 0.3) | 0.98 | −0.0 (−0.2 to 0.2) | 0.91 | 96 | −0.5 (−1.6 to 0.5) | 0.31 | −0.3 (−1.1 to 0.5) | 0.40 | 67 | −0.2 (−1.6 to 1.2) | 0.82 | −0.1 (−1.1 to 0.9) | 0.83 |
| Sleep duration | 90 | −0.1 (−0.3 to 0.1) | 0.36 | −0.0 (−0.2 to 0.2) | 0.98 | 90 | −0.5 (−1.4 to 0.4) | 0.26 | −0.1 (−0.9 to 0.7) | 0.81 | 62 | −0.1 (−1.6 to 1.3) | 0.88 | 0.3 (−0.8 to 1.5) | 0.54 |
| **Model 2** | | | | | | | | | | | | | | | |
| Sleep onset | 96 | 0.0 (−0.2 to 0.3) | 0.87 | −0.1 (−0.4 to 0.1) | 0.38 | 96 | 0.1 (−0.9 to 1.0) | 0.90 | −0.0 (−1.0 to 0.9) | 0.92 | 67 | −0.3 (−1.6 to 1.0) | 0.68 | −0.9 (−2.1 to 0.2) | 0.12 |
| Sleep period | 96 | 0.1 (−0.2 to 0.4) | 0.66 | 0.0 (−0.2 to 0.2) | 0.81 | 96 | −0.4 (−1.5 to 0.8) | 0.51 | −0.3 (−1.1 to 0.6) | 0.54 | 67 | −0.1 (−1.6 to 1.3) | 0.86 | 0.1 (−1.0 to 1.1) | 0.89 |
| Sleep duration | 90 | −0.1 (−0.3 to 0.2) | 0.55 | 0.1 (−0.2 to 0.3) | 0.55 | 90 | −0.4 (−1.3 to 0.5) | 0.37 | 0.1 (−0.8 to 1.0) | 0.81 | 62 | −0.2 (−1.7 to 1.3) | 0.79 | 0.5 (−0.8 to 1.8) | 0.44 |
| **Model 3** | | | | | | | | | | | | | | | |
| Sleep onset* | 96 | 0.0 (−0.2 to 0.3) | 0.79 | −0.1 (−0.4 to 0.1) | 0.33 | 96 | 0.0 (−1.0 to 1.0) | 0.98 | −0.2 (−1.3 to 0.9) | 0.71 | 67 | −0.3 (−1.6 to 1.0) | 0.67 | −1.3 (−2.8 to 0.1) | 0.074 |
| Sleep onset† | 90 | 0.0 (−0.3 to 0.3) | 0.98 | −0.0 (−0.3 to 0.2) | 0.72 | 90 | 0.0 (−1.0 to 1.0) | 0.98 | 0.1 (−0.9 to 1.1) | 0.83 | 62 | −0.4 (−1.9 to 1.2) | 0.65 | −1.4 (−3.1 to 0.2) | 0.085 |
| Sleep period | 96 | 0.1 (−0.2 to 0.4) | 0.63 | −0.0 (−0.3 to 0.2) | 0.76 | 96 | −0.4 (−1.6 to 0.8) | 0.52 | −0.3 (−1.3 to 0.6) | 0.48 | 67 | −0.2 (−1.6 to 1.3) | 0.83 | −0.6 (−2.0 to 0.7) | 0.33 |
| Sleep duration | 90 | −0.1 (−0.3 to 0.2) | 0.55 | 0.1 (−0.2 to 0.3) | 0.68 | 90 | −0.4 (−1.4 to 0.5) | 0.37 | 0.1 (−0.8 to 1.1) | 0.76 | 62 | −0.3 (−1.8 to 1.3) | 0.74 | −0.3 (−2.0 to 1.3) | 0.67 |

Results are beta coefficients (95% CIs) and represent the difference in outcomes for every 1 hour later to sleep, 1 hour longer sleep period or duration, 1 hour later to sleep on weekends than weekdays, or 1 hour longer sleep period or duration on weekends than weekdays. Model 1 was adjusted for sex, age, socioeconomic status, maternal pregnancy age, parity, gestational age, birth weight, maternal body mass index, season of measurement and napping frequency (models for weekday to weekend differences were further adjusted for the average weekday value of the exposure variable). Model 2 was further adjusted for unhealthy snacking, fruit and vegetable intake, and TV viewing (models for weekday to weekend sleep differences were adjusted for average weekday TV viewing and weekday to weekend differences in TV viewing). Model 3 further adjusted sleep period and duration variables for sleep onset, and vice versa (models for weekday to weekend differences in the sleep period, duration or onset as appropriate). Statistically significant results (p<0.05) are highlighted bold. Sensitivity analyses confirmed that adjustment for physical activity (collected at 36 months) did not confound model 3 results, but the association between weighted daily average sleep onset with body mass index z-score at 18 months was attenuated when adjusted for time of last meal (body mass index z-score (n=101): −0.2 (−0.5 to 0.2), p=0.28).
*Adjusted for the sleep period.
†Adjusted for sleep duration.

around 16:00 and tails-off by 20:00.[42 43] In our sample of white children aged 18 months, the earliest sleep onset times were around 18:00 and more than one-third (36.8%) were asleep by 19:30. It is conceivable that these children may have missed a substantial part of the evening peak in physical activity, thus contributing to higher adiposity.[33 44] Our related observation in the same group of children, that going to sleep ≥20 min later on weekends than weekdays was associated with lower waist circumference, may be explained by later sleep onset at the weekend enabling more opportunity to be physically active.[45]

Although white children aged 36 months exhibited a similar distribution of sleep onset times compared with their younger white peers, there were few comparable associations, apart from some indication in children aged 36 months that later sleep onset on weekends than weekdays was associated with lower adiposity (an association with the sum of two-skinfolds approached statistical significance). The apparent age-related differences may be explained by daily physical activity patterns following a different trajectory in slightly older children who tend not to nap (just 5% of white children aged 36 months napped every day; 46% napped occasionally; 49% did not nap). Instead of a naptime-related slump after mid-day which is followed by an evening peak, physical activity levels tend to be relatively constant throughout the afternoon, before progressively declining in the evening and most notably from 7pm onwards.[46] Hence, in many white children aged 36 months, there may have been no peak in evening physical activity for early sleep to obstruct. In contrast to our results, Roy *et al* did find a significant association between later bedtimes with lower BMI z-score in children aged approximately 36 months.[20] That study was performed in children living in America, Australia and New Zealand, which may account for the discrepancy. In those countries frequent daytime naps persist in older children partly because of designated naptimes in childcare.[47–49] Hence, many children aged 3–4 years continue to exhibit a bimodal physical activity pattern, including an early-to-mid evening activity peak, which early sleep might obstruct.[50]

Sleep recommendations for children have historically focused on duration rather than other sleep dimensions.[51] Our results highlight that contemporary sleep guidelines should further acknowledge the importance of sleep timing and regular sleep schedules. Any such recommendations should also be considerate of ethnic and cultural differences in sleeping patterns, including what may and may not be feasible and acceptable to different population subgroups. Our results for white children aged 18 months, some of whom were already asleep just after 18:00, indicate that going to sleep too early could be detrimental from an energy-balance and obesity risk perspective. Sleep onset times were much later in South Asian children, who may benefit from an earlier evening meal, earlier sleep onset, and consistent sleep schedules across week and weekend days. It is difficult to decipher if any such changes would translate to clinically meaningful differences in adiposity levels because young children tend to be healthy. However, more than half (54%) of South Asian children aged 18 months went to sleep later on weekends than weekdays, on average by 32 min (±23 min). According to our point estimates, this could be associated with up to 1 cm larger waist circumference. This is unlikely trivial considering that any beneficial influence on adiposity levels this early in childhood may be important. It is a critical period when rapid weight gain occurs and sustained obesity can develop.[18]

This study uniquely investigated associations of diarised sleep parameters with myriad adiposity markers in a biethnic sample of young children from a deprived urban setting, which is a high-risk group for sleep problems and obesity.[22 23] Sleep diaries enabled information about numerous dimensions of sleep to be collated but unfortunately diaries incur participant burden. This likely contributed to only a minority of BiB 1000 parents choosing to complete sleep diaries for their child, the consequences of which are at least twofold. First, our study sample is subject to selection biases. Compared with the full BiB pregnancy cohort, which was broadly representative of the obstetric population in Bradford at the time of recruitment, the proportion of families who were most deprived was lower in this study (28% vs 45%).[34] This hinders generalisability, but it is reassuring that there was still variability within the sample in terms of parental socioeconomic status. Second, stratifying by ethnicity and age group meant that some of our analyses were based on relatively few observations and low statistical power. This latter issue was compounded by missing data for specific items such as sleep duration (which was only reported if parents considered they knew how long their child had slept for overnight) and the sum of two-skinfolds. The results should, therefore, be considered exploratory and require replication in larger samples. It is advantageous that we investigated more direct adiposity markers than weight-for-height proxies, which tend to underestimate fatness in South Asians,[52] and that in addition to skinfolds we investigated waist circumferences, as South Asian children eventually develop more centrally stored adiposity which is metabolically deleterious.[26] Another strength is that each of our analyses were adjusted for a broad range of covariates, including mutual adjustment for sleep parameters, to tease apart the independent associations of distinct sleep dimensions with adiposity. It is a weakness that we did not have information about napping durations, which would have permitted investigation of napping and 24-hour sleeping patterns with outcomes. Furthermore, directions of association cannot be inferred from this cross-sectional study since it is possible that adiposity could influence sleeping habits and vice versa.[24] Longitudinal studies or trials are needed to determine directions of association. Future studies may achieve a better trade-off between participant retention and data resolution if they use sensor-based technologies rather than diaries,[53] although a combination of methods may prove best to capture habitual sleep parameters.[54]

## CONCLUSIONS

Sleep onset times appear to be independently associated with levels of total and central adiposity in young South Asian and white children. Suitable sleep times and consistent sleep schedules across weekdays and weekend days may be important modifiable determinants of early childhood obesity.

**Contributors** JW is chief investigator of the Born in Bradford study and setup the cohort. HLB created the parent-reported sleep diary. PJC initiated the study, planned and performed the statistical analyses, and wrote the manuscript. All authors including JEB and EP critically revised the manuscript for intellectual content, helped interpret study findings, read and approved the final version and are accountable for the work.

**Funding** The Born in Bradford study receives core infrastructure funding from the Wellcome Trust (grant number: WT101597MA), a joint grant from the UK Medical Research Council (MRC) and UK Economic and Social Science Research Council (ESRC) (grant number: MR/N024397/1), the British Heart Foundation (BHF) (grant number: CS/16/4/32482) and the National Institute for Health Research (NIHR) under its Collaboration for Applied Health Research and Care (CLAHRC) for Yorkshire and Humber. This study received delivery support from the NIHR Clinical Research Network. JW leads the Healthy Children, Healthy Families Theme of the NIHR CLAHRC in Yorkshire and Humber. PJC is funded by a BHF Immediate Postdoctoral Basic Science Research Fellowship (grant number: FS/17/37/32937). JEB coordinates and PJC is a member of The White Rose Child and Adolescent Sleep Research Network, which is funded by a White Rose Collaboration Grant. The views expressed in this paper are those of the authors and not necessarily those of the MRC, ESRC, BHF, NIHR and UK Department of Health or National Health Services or of any other funder acknowledged here.

**Competing interests** None declared.

**Patient and public involvement** Patients and/or the public were not involved in the design, or conduct, or reporting, or dissemination plans of this research.

**Patient consent for publication** Not required.

**Ethics approval** Ethical approval for all aspects of the study was granted by Bradford Research Ethics Committee (Reference 07/H1302/112) and all mothers provided informed written consent for participation.

**Provenance and peer review** Not commissioned; externally peer reviewed.

**Data availability statement** Data are available upon reasonable request.

**ORCID iDs**
Paul James Collings http://orcid.org/0000-0003-2022-5453
John Wright http://orcid.org/0000-0001-9572-7293

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
