## [Reviewer comments · BMJ Open]

ARTICLE DETAILS

TITLE (PROVISIONAL)	Associations of diarised sleep onset time, period, and duration with total and central adiposity in a biethnic sample of young children: the Born in Bradford observational cohort study
AUTHORS	Collings, Paul; Blackwell, Jane; Pal, Elizabeth; Ball, Helen; Wright, John

VERSION 1 – REVIEW

REVIEWER	Professor Rachael Taylor University of Otago, New Zealand
REVIEW RETURNED	05-Oct-2020

GENERAL COMMENTS	The authors present an interesting analysis of the cross-sectional relationships between sleep duration and timing with anthropometric measures of adiposity in a small group of young children measured at two time points - 18 and 36 months of age. While a reasonably large body of literature exists that has examined the relationships between sleep duration and obesity, and to a lesser extent sleep timing and obesity, in children, relatively few papers exist in younger children, making this a welcome addition to the literature. However, I have a few queries and suggestions that I believe would strengthen the manuscript as follows: 1. Given the dearth of longitudinal studies examining the relationships between sleep and obesity, it seems surprising that the authors have chosen to only undertake cross-sectional analyses at the two time-points - I can only assume that these do not represent the same children, rather, individual children who had complete measures at either time point. Are any longitudinal analyses possible or do much smaller numbers preclude this?2. Tables 2 and 3 show a very large number of analyses with very few significant findings - I would argue perhaps little more than would be expected by chance alone. How confident can the authors be that their "significant" findings are genuine observations, particularly perhaps given the inconsistency in results including across ethnicity? I think the paper should be reworded throughout to "tone down" the findings somewhat in light of this.3. Similarly, greater attention should be paid to the "clinical" rather than statistical significance of the findings. This is very briefly mentioned at the end of the discussion but not in sufficient detail I believe. The author's assertion that parents of White children should be guided to send their children to bed after 7.30pm on the basis of a single significant finding (with many others not significant) is quite a stretch.4. Little attention has been paid to napping and how this might have impacted the findings. Just collecting whether a child had a nap or not provides little indication of 24-hour sleeping patterns which
---

	would be a more important variable to have in this age group. 5. Introduction 3rd paragraph - I think this is slightly misleading - only one study might have examined sleep duration AND bedtime with adiposity in the same paper - but there is a larger body of literature that has examined EITHER bedtime or duration with adiposity. The introduction should be revised to reflect this. 6. This group represents a very small fraction of the larger BiB sample - there must be plenty of data to illustrate how this group differed from the larger sample - and how this might impact interpretation of the findings. 7. Although 3 days of sleep diary was good to get, how confident are the authors that a single night of weekend sleep (or two for weekday) truly reflect "usual" timings? They assert that it is much better than asking about "usual" at this age but do we actually know this is true? 8. There is some argument over the appropriateness of adjusting sleep timing for sleep duration and vice versa given the very close relationship between the two - is there any chance of multicollinearity here? 9. I was curious with "last mealtime" as a confounder and wondered whether included milk feeds or just solid food? 10. Page 13, line 39 - do UK childcare centres not have prescribed napping periods? 11. Please provide references to support the claim about direct adiposity measures (page 14) - do we really know that waist circumference or skinfolds are a "better" measure of adiposity in 18 month olds? Or 36 month olds?
--	---

REVIEWER	Jian Guan Shanghai 6th Peoples Hospital Affiliated to Shanghai Jiaotong University School of Medicine, Department of Otolaryngology Head and Neck Surgery & Center of Sleep Medicine
REVIEW RETURNED	13-Jan-2021

GENERAL COMMENTS	1. 3-day sleep diary was used to record the sleep quality of children aged 18 and 36m at night, the sleep time in the daytime was ignored, a standard polysomnography may be a better tool to record children's daily sleep onset time, period and duration. 2. What is the definition of central adiposity especially in children aged 18 and 36m. 3. In the of table 2, model 2 was adjusted for unhealthy snacking, fruit and vegetable intake, there is no description of the food intake in the manuscript and in table1. 4. "We discovered ethnicity- and age-specific associations for sleep onset times, which we primarily attribute to markedly different sleep schedules between South Asian and White children" (P11 line5-8) could not be drawn from the results of the manuscript. 5. The sample was small. 3-day sleep diary may not represent the sleep onset time, period and duration of children.
---

VERSION 1 – AUTHOR RESPONSE

Reviewer 1:

Dr. Rachael Taylor, University of Otago

Comments to the Author:

The authors present an interesting analysis of the cross-sectional relationships between sleep

duration and timing with anthropometric measures of adiposity in a small group of young children measured at two time points - 18 and 36 months of age. While a reasonably large body of literature exists that has examined the relationships between sleep duration and obesity, and to a lesser extent sleep timing and obesity, in children, relatively few papers exist in younger children, making this a welcome addition to the literature.

Thank you for taking the time to review our manuscript and for providing helpful comments. We hope that all issues have been addressed to your satisfaction.

However, I have a few queries and suggestions that I believe would strengthen the manuscript as follows:

1. Given the dearth of longitudinal studies examining the relationships between sleep and obesity, it seems surprising that the authors have chosen to only undertake cross-sectional analyses at the two time-points - I can only assume that these do not represent the same children, rather, individual children who had complete measures at either time point. Are any longitudinal analyses possible or do much smaller numbers preclude this?

We had hoped to perform a longitudinal analysis but only up to 84 children ($n=25$ South Asian; $n=59$ White) had complete data at the 18 and 36 month timepoints. In the future it will be possible to investigate associations of sleep diary data with adiposity trajectories – we have weight, height and skinfold measures collected over time in the BiB cohort – but it will not be possible to factor into the analysis or interpretation of results changes in sleep habits or covariates over time. This is one of very few studies on the topic of myriad sleep parameters and adiposity in young children. We also uniquely present ethnicity-specific associations. Although cross-sectional, we agree that this is a worthwhile contribution to the literature.

2. Tables 2 and 3 show a very large number of analyses with very few significant findings - I would argue perhaps little more than would be expected by chance alone. How confident can the authors be that their "significant" findings are genuine observations, particularly perhaps given the inconsistency in results including across ethnicity? I think the paper should be reworded throughout to "tone down" the findings somewhat in light of this.

This is the first study to quantify associations of sleep parameters with direct markers of adiposity in this age group and to investigate differences of associations by ethnicity. The analysis could be considered exploratory – which would be one reason not to adjust the threshold for statistical significance to a more conservative level to offset chance findings. Furthermore, the various hypothesis tests that have been performed are not 100% independent of each other. Tables 2 and 3 report results from different levels of adjustment for the same exposure-outcome pairs (models 1-3) and each of the sleep parameters and adiposity markers were correlated to various degrees (as we describe in the first paragraph of the results section). Using a Bonferroni correction factor, for example, would have resulted in overly conservative adjustments to the alpha-level.

3. Similarly, greater attention should be paid to the "clinical" rather than statistical significance of the

findings. This is very briefly mentioned at the end of the discussion but not in sufficient detail I believe. The author's assertion that parents of White children should be guided to send their children to bed after 7.30pm on the basis of a single significant finding (with many others not significant) is quite a stretch.

Considering your comments we have toned down our recommendations. It is difficult to make judgements as to what may or may not constitute clinically meaningful associations in young children who are generally healthy. Judgements such as these would have been easier if overweight and obesity were also specified as outcomes – but deriving weight status categories would have created few cases in some already small strata. It would have also been desirable to apply BMI adjustments before deriving weight status categories, to account for ethnic differences in the BMI-body fatness association (BMI tends to underestimate body fatness in South Asian children), but unfortunately such BMI adjustments are only currently available for children aged 4y and older. We have reflected on the potential clinical significance of our findings and have edited the discussion accordingly:

It is difficult to decipher if any such changes would translate to clinically meaningful differences in adiposity levels because young children tend to be healthy. However, more than half (54%) of South Asian children aged 18m went to sleep later on weekends than weekdays, on average by 32 (\pm 23) minutes. According to our point estimates this could be associated with up to 1cm larger waist circumference. This is unlikely trivial, particularly considering that any beneficial influence on adiposity levels this early in childhood may be important, as it is a critical period when rapid weight gain occurs and sustained obesity can develop [18].

4. Little attention has been paid to napping and how this might have impacted the findings. Just collecting whether a child had a nap or not provides little indication of 24-hour sleeping patterns which would be a more important variable to have in this age group.

Unfortunately we had little information about napping other than whether children napped on sleep diary reporting days. We agree that sleeping patterns across a whole 24h day are important to investigate in young children. However, we also believe that it is important to differentiate between napping and overnight sleep, they are two distinct periods which might conceivably exhibit different associations with health. The focus of this study was on night time sleep.

5. Introduction 3rd paragraph - I think this is slightly misleading - only one study might have examined sleep duration AND bedtime with adiposity in the same paper - but there is a larger body of literature that has examined EITHER bedtime or duration with adiposity. The introduction should be revised to reflect this.

We apologise if this was unintentionally misleading. Since we submitted this paper another study has been published on associations of sleep timing consistency with early childhood adiposity. We refer to this study in the discussion, and in the introduction we now highlight that whilst an increasing number of studies have investigated sleep duration and adiposity in young children, few have investigated timing or variability.

6. This group represents a very small fraction of the larger BiB sample - there must be plenty of data to illustrate how this group differed from the larger sample - and how this might impact interpretation of the findings.

The study sample is likely subject to selection biases, as we note in 'Strengths and limitations' set of bullet points, and the concluding paragraph of the discussion. Compared to the full BiB pregnancy cohort the sleep diary subsample included a lower proportion of the 'most deprived' families. We have added this information to the discussion section. We don't perceive that this would have impacted the study results, which were adjusted for numerous factors including parental socioeconomic status. It does hinder generalisability somewhat but reassuringly there was still variability in parental socioeconomic status and other descriptors.

7. Although 3 days of sleep diary was good to get, how confident are the authors that a single night of weekend sleep (or two for weekday) truly reflect "usual" timings? They assert that it is much better than asking about "usual" at this age but do we actually know this is true?

Sleep diaries are advantageous because they document real-time behaviours thereby removing issues related to recall bias. In the final paragraph of the discussion we were not attempting to convey that enquiring about 'usual' sleep behaviours is inferior, rather that the common practice of posing a single question to parents about their child's sleep provides limited information. We have edited these sentences accordingly. Sleep is a multidimensional construct and by using sleep diaries we were able to collate valuable information about numerous dimensions of sleep. We comment on the adequacy of 3 days of diary data in our response to the final comment made by Reviewer 2 (please see below).

8. There is some argument over the appropriateness of adjusting sleep timing for sleep duration and vice versa given the very close relationship between the two - is there any chance of multicollinearity here?

We believe that this is a bigger issue in school children for whom awake times are often very similar because of set school start times; consequently sleep onset times and duration can be very closely correlated. There is likely more variability in young children's awake times. Young children also tend to wake more overnight which would likely weaken correlations. In the first paragraph of the results section we report correlations between sleep onset time, period and duration variables: the largest correlations were only moderate (-0.52). As part of the main analyses we calculated post-regression variance inflation factors, which were all within acceptable limits. We have added this final detail to the results section.

9. I was curious with "last mealtime" as a confounder and wondered whether included milk feeds or just solid food?

Parents provided an answer to the question 'What time did your child have their last meal today?' No guidance was given regarding what constituted a meal.

10. Page 13, line 39 - do UK childcare centres not have prescribed napping periods?

Many young children in the UK do not nap whilst at nursery, either because there is no dedicated sleep area or because the parent has specifically asked for their child not to be given a nap.

11. Please provide references to support the claim about direct adiposity measures (page 14) - do we really know that waist circumference or skinfolds are a "better" measure of adiposity in 18 month olds? Or 36 month olds?

While BMI remains a widely accepted surrogate measure of adiposity its limitations are well documented, including an inability to discriminate between fat and lean tissue. It is also a measure that has different diagnostic performance in some ethnic groups. For example, BMI tends to underestimate body fatness in South Asian children. Given the composition of our study sample it is advantageous that we were also able to investigate additional markers of adiposity. We think that including waist circumference as an outcome was particularly important, as South Asians accumulate more centrally-stored adiposity which is known to be metabolically harmful. We have made some edits in order to clarify our position.

Reviewer 2:

Dr. Jian Guan, Shanghai 6th Peoples Hospital Affiliated to Shanghai Jiaotong University School of Medicine

Comments to the Author:

1. 3-day sleep diary was used to record the sleep quality of children aged 18 and 36m at night, the sleep time in the daytime was ignored, a standard polysomnography may be a better tool to record children's daily sleep onset time, period and duration.

Thank you for reviewing our manuscript. We had no information about sleep quality but we did investigate associations of sleep onset time, period, duration and variability with adiposity markers. It was not ignored but we do acknowledge that we had limited information about napping. As per a response to Reviewer 1, we believe that it is important to differentiate between napping and overnight sleep, they are two distinct periods which might conceivably have different associations with health. The focus of this study was on night time sleeping. We accept that polysomnography is considered a gold-standard measure of sleeping behaviours. However, it is an obtrusive technique and can affect normal sleeping patterns and behaviour. From our experience recruiting families to participate in polysomnography measures with their young children is also extremely challenging.

2. What is the definition of central adiposity especially in children aged 18 and 36m.

We considered waist circumference measured at the level of the navel to be indicative of central adiposity – as have other studies of similarly aged children. We modelled waist circumference continuously and imposed no thresholds to identify 'large' waist circumferences. As mentioned above in reply to Reviewer 1, South Asians have more centrally-stored adiposity which is metabolically deleterious. Given the ethnic composition of our study sample waist circumference was considered to

be an important outcome. It is advantageous that we were able to study associations with three outcomes. Correlations between the various adiposity markers (reported in the first paragraph of the results section) indicate that they all provided different measures of adiposity.

3. In the footnote of table 2, model 2 was adjusted for unhealthy snacking, fruit and vegetable intake, there is no description of the food intake in the manuscript and in table 1.

It is unfortunate that you missed this. In the methods section we described measurement of diet variables and in Table 1 we summarised data for both unhealthy snacks (*n*/week) and fruit & vegetables (portions/day).

'Infant dietary data were collected using a validated food frequency questionnaire that was modified to include ethnic-specific foodstuffs. Dietary constructs indicative of unhealthy snacking (frequency of biscuit, crisps, cakes, sweets, chocolate, sugar-sweetened beverage consumption) and fruit and vegetable consumption in the previous 4-12 weeks were derived [24].'

4. "We discovered ethnicity- and age-specific associations for sleep onset times, which we primarily attribute to markedly different sleep schedules between South Asian and White children" (P11 line5-8) could not be drawn from the results of the manuscript.

We have edited this sentence to provide clarity of meaning. Rather than simply report associations and fail to provide any conceivable justification for them (which unfortunately is common the literature) we have made a concerted effort to provide plausible explanations as to why our data show what they show. Later daily sleep onset, and later sleep onset on weekends than weekdays, were both associated with higher adiposity in South Asian children and conversely lower adiposity in White children. We consider it very reasonable to hypothesise that markedly different sleep schedules, specifically South Asian children going to sleep on average nearly 2h later than White children, likely underpin the differences of associations.

5. The sample was small. 3-day sleep diary may not represent the sleep onset time, period and duration of children.

Relative to other studies the total sample size was not particularly small. Stratifying the analyses by age group and ethnicity, which we considered to be important, did however produce smaller strata. Sleep diaries allowed us to capture information about numerous dimensions of sleep but we acknowledge in the discussion that alternative, possibly mixed-methods, may have been preferable:

'Future studies may achieve a better trade-off between participant retention and data resolution if they use sensor-based technologies rather than diaries [54] although a combination of methods may prove best to capture habitual sleep parameters [55].'

With regards to the issue of whether 3 days of data are satisfactory or not. It is always beneficial to have more data, but there is a trade-off between data resolution and participant burden. It is possible that asking parents to complete diaries for every night of an entire week or less frequently over the course of a fortnight would have resulted in fewer diaries being completed, leading to a smaller and

more biased sample. Rightly or wrongly, actigraphy-based sleep estimates have set a precedent for the amount of information that qualifies for an acceptable observation period. That is typically ≥ 3 nights of data including one weekend night.

VERSION 2 – REVIEW

REVIEWER	Professor Rachael Taylor University of Otago, New Zealand
REVIEW RETURNED	28-Jan-2021

GENERAL COMMENTS	The authors have addressed almost all of my concerns or queries to my satisfaction. I have just two minor points: Original point 4 re napping. I acknowledge that this paper is about night-time sleeping but at this age, how long the child naps for will influence their night-time sleep. The fact that you only had yes/no for naps and not duration needs to be added as a limitation. Original point 8. Please add in the VIF calculated.
--

VERSION 2 – AUTHOR RESPONSE

Reviewer 1:

Dr. Rachael Taylor, University of Otago

Comments to the Author: The authors have addressed almost all of my concerns or queries to my satisfaction.

Thanks once more for taking the time to review our manuscript.

I have just two minor points:

1. Original point 4 re napping. I acknowledge that this paper is about night-time sleeping but at this age, how long the child naps for will influence their night-time sleep. The fact that you only had yes/no for naps and not duration needs to be added as a limitation.

We have now included the following sentence: 'It is a weakness that we did not have information about napping durations which would have permitted investigation of napping and 24-hour sleeping patterns with outcomes.'

2. Original point 8. Please add in the VIF calculated.

All variance inflation factors were below 10 and the majority were below 2.5. We have added this information to the manuscript.